# Translating Guidelines into Practice: A Multicentre Audit of the Implementation of ERC Survivorship and Follow-Up Recommendations After Cardiac Arrest

**DOI:** 10.3390/jcm15010174

**Published:** 2025-12-25

**Authors:** Marco Mion, Meadbh Keenan, Alice Steadman, Shirley Morrison, Claudine Keelan, Nikos Gorgoraptis, Nilesh Pareek, Jean Davis, Uzma Sajjad, Thomas R. Keeble

**Affiliations:** 1Essex Cardiothoracic Centre, Mid and South Essex NHS Foundation Trust, Basildon SS16 5NL, UK; 2Medical Technology Research Centre, Anglia Ruskin School of Medicine, Chelmsford CB1 1PT, UK; 3King’s College Hospital, NHS Foundation Trust, London SW3 6PY, UK; alice.steadman1@nhs.net; 4Barts Heart and Thorax Centre, St. Bartholomew’s Hospital, London EC1A 7BE, UK; meadbh.keenan@nhs.net; 5Norfolk and Norwich University Hospitals NHS Foundation Trust (NNUHFT), Norwich NR4 7UY, UK; 6Barts Health NHS Trust, London E1 1BB, UK; 7School of Cardiovascular Metabolic Medicine & Sciences, BHF Centre of Excellence, King’s College London, London SE5 8AF, UK

**Keywords:** out-of-hospital cardiac arrest, clinical audit, guideline adherence, aftercare, implementation science

## Abstract

**Background**: Survivors of sudden cardiac arrest frequently experience long-lasting problems with fatigue, cognition and mood. European Resuscitation Council (ERC) guidelines recommend functional assessment of physical/non-physical issues prior to discharge, and systematic review within three months covering at least cognition, mood, fatigue, and support for patients and their families. How these recommendations are implemented and what barriers are encountered in routine care remains unknown. **Methods**: We conducted a multicentric, prospective 6-month audit across four tertiary cardiac-arrest centres in England where temporarily funded follow-up pathways were in place. Five operational criteria were developed based on ERC guidelines. Adherence was quantified, and reasons for non-completion were collected and mapped onto the Theoretical Domains Framework (TDF) to identify behavioural and contextual factors influencing implementation. **Results**: A total of 143 OHCA survivors were discharged alive. Pre-discharge assessments were offered to 116/143 patients (81%) but only completed in 81 (57%). Reasons for non-completion included early discharge, severe cognitive impairment and, less frequently, patient refusal. Of 132 survivors eligible for follow-up, 108 (82%) were contacted and 87 (66%) attended. Only 25% of follow-ups occurred within the recommended 3-month period (median 185 days, IQR 81–225). Among those seen, assessments were completed for cognition (44%), mood (52%), and fatigue (51%). Reasons for omission included patient refusal, clinical discretion, and time constraints. Survivors’ family members were invited in all cases, but only 45% attended. **Conclusions**: Adherence to guideline-recommended assessments was variable and dependent on local practices, resource limitations, and patient/clinician-related factors. Key barriers mapped to the TDF domains of ‘Environmental context and resources’, ‘Beliefs about consequences’ and ‘Social influences’. Structural policies excluding out-of-area and non-ICU patients, together with clinician judgement and patient engagement, were major determinants of implementation. These findings can guide targeted service development and support sustainable post-resuscitation care pathways.

## 1. Introduction

Sudden cardiac arrest affects around 30,000 people every year in the UK. Of the 8–10% of patients who survive this event, around 70% experience fatigue, half have some degree of cognitive impairment and between 1/4 and 1/3 experience significant symptoms of low mood, anxiety and/or post-traumatic stress disorder six months after the event [1]. Recent guidelines [1] position statements [2,3] and quality standards [4] recommend the investigation of all these domains, before and after discharge. To date, however, no study has been published on the implementation of these guidelines in routine clinical practice.

Guidance on how to complete the screening is scant. European Resuscitation Council guidelines recommend performing “functional assessments of physical and non-physical impairment” before discharge from hospital: this refers to a wide range of (semi)structured evaluations aimed at understanding the patient’s current capabilities and limitations, and to inform rehabilitation, discharge planning, and ongoing care. Their use has been relatively limited in published studies, with a few exception—namely, the Frenchay Activities Index (FI) [5], the Katz Index of Independence in Activities of Daily Living [6], the assessment of Motor and Process Skills (AMPS) and the Activities of Daily Living Interview (ADL-I) [7,8] which, however, are unlikely to be widely used in routine clinical practice where the assessment of ADLs is more often unstandardised, variable and informal.

ERC guidelines recommend using the MoCA (Montreal Cognitive Assessment) to formally evaluate cognition, and the IQCODE-CA (Informant Questionnaire of Cognitive decline in the Elderly—Cardiac Arrest version) and the CLCH-24 (Checklist Cognition and Emotion) to investigate informants’ and patients’ insight into cognition and behaviour, respectively, if required.

The HADS (Hospital Anxiety and Depression Scale) is also recommended for the screening of emotional problems and widely available. No specifically validated tool is recommended to assess fatigue, despite this being the most common complaint, with only a handful of tools having been used in the literature (Modified Fatigue Impact Scale; Fatigue Severity Scale; Patient-Reported Outcome Measurement Information System-Fatigue Scale; Multidimensional Fatigue Inventory (MFI-20) [9,10,11]).

In this context, working practices in the use of screening tools for cognition, mood and fatigue are largely understudied in the UK; unsurprisingly, no information is currently available on what tools are being used in different cardiac arrest centres (as defined in [12]) to screen survivors following an OHCA, their acceptability to patients and what type of barriers and facilitators can affect their routine use.

In this audit, we operationalised the 2021 ERC–ESICM post-resuscitation guidelines into five criteria and prospectively audited adherence across four centres in South-East England. The 2025 guidelines update preserves the same core elements (functional assessment before discharge; organised ≤3-month follow-up with cognitive, emotional, and fatigue screening; information/support for patients and co-survivors [13], so our endpoints remain aligned; the added emphasis on structured rehabilitation does not change them.

## 2. Methods

### 2.1. Structure of the Clinical Audit

#### 2.1.1. Participating Organizations and Timeframe of Audit

A multidisciplinary working group on ‘care and rehabilitation’ following OHCA first met in December 2022, comprising therapists from 4 different NHS Hospital Trusts in Southeast/Eastern England (King’s College Hospital, NHS Foundation Trust; Barts Health NHS Trust; Mid and South Essex NHS Foundation Trust; Norfolk and Norwich University Hospital NHS Foundation Trust). These specialized tertiary hospital units serve a large catchment area across London, Essex and Norfolk, covering a population of around 6.2 million people, and at the time had established pilot systems to complete pre-discharge and follow-up assessments for OHCA survivors (Figure 1).

To gain some system-level insights into the way these systems were structured and implemented, the group decided to engage in a prospective, multi-centre, 6-month audit of current practices in the assessment of cognition, mood, fatigue, and quality of life following an OHCA.

The audit was divided into pre-discharge and follow-up sections. All patients admitted between the 1 June 2023 and the 30 November 2023 who were discharged alive from hospital were included in the audit; the follow-up section of the audit was completed when the last patient recruited was seen at follow-up.

#### 2.1.2. Data Collection Methods and Tools Used

A single audit proforma and accompanying guidance document were developed by a multidisciplinary working group, based on the 2021 ERC post-resuscitation care guidelines. At each site, a designated clinical lead (specialist occupational therapist or clinical psychologist) extracted data from electronic health records and local cardiac arrest databases into the shared proforma (Appendix A). Operational definitions of pre- and post-discharge criteria were specified for each item (see Audit Criteria and Standards Used Section 2.1.3). The coordinating centre provided clarification as needed via email and online meetings. The final anonymised dataset was collated centrally and subjected to range and logic checks (e.g., consistency between discharge status and follow-up entries, plausible date sequences); any discrepancies were resolved by re-checking source records at the relevant site. Details of specific cognitive assessments and questionnaires used in each centre are provided in Appendix A.

#### 2.1.3. Audit Criteria and Standards Used

This audit assessed the compliance with European Resuscitation Council guidelines on post-resuscitation care. Two standards were specifically investigated:

Standard 1—pre-discharge care “*Providing information and performing functional assessments of physical and non-physical impairments before discharge from the hospital*”.

Standard 2—post-discharge care “*Systematic follow-up of all cardiac arrest survivors within 3 months following hospital discharge, which should, at least, include cognitive screening, screening for emotional problems and fatigue, and the provision of information and support for patients and their family*”. Based on these standards, we derived and operationalised five audit criteria with explicit definitions and denominators to ensure reproducible coding (Table 1).

For the purpose of this audit, a pre-discharge ‘functional assessment’ was defined as a structured or semi-structured, performance-based evaluation of everyday activities, capturing both physical functioning and *functional cognition*—that is, how cognitive skills are applied to real-world tasks. It was coded as completed if, before discharge, the patient underwent at least one functional assessment listed in the proforma (e.g., Multiple Errands Test, PADL assessment, etc.) and this was documented in the medical record. Isolated cognitive screening (e.g., MoCA, ACE-III, FreeCog), general mobility grading (e.g., transfers or walking with/without aids), narrative observations from ward staff, and patient self-report of independence did not fulfil this criterion. *Provision of information* was defined as documented provision of written or verbal information about cognitive, emotional, and fatigue sequelae.

#### 2.1.4. Identification and Coding of Barriers and Facilitators

For each audit criterion, local leads categorised reasons for non-completion using a list of predefined options (e.g., patient fatigue, competing clinical priorities, lack of staff time, perceived lack of necessity) and provided free-text comments where needed. These reasons were then mapped to the 14-domain Theoretical Domains Framework (TDF) by a clinical psychologist (MM) with experience in behaviour change and implementation science, to help identify mechanisms of change within complex healthcare settings [14].

Coding was reviewed and refined in team meetings, and any uncertainties resolved by consensus. When a barrier clearly spanned more than one TDF domain, the domain judged most salient was selected as the primary code.

#### 2.1.5. Ethics

According to guidance published by the Healthcare Quality Improvement Partnership, this project was classified as a clinical audit and registered at MSE (Mid and South Essex NHS Trust) with reference number CTCCA103 in October 2023. No ethics application was required as patients continued to receive standard clinical care without any modification or intervention beyond routine practice.

#### 2.1.6. Data Analysis and Interpretation

Descriptive statistics were used to characterize patient demographics, clinical features, and the frequency of assessments completed pre-discharge and during follow-up. Continuous variables (e.g., age, length of hospital stay, days from hospital discharge to follow-up) were summarized using medians and interquartile ranges (IQR), reflecting the non-normal distribution typically seen in clinical audit data. Categorical variables (e.g., sex, location of arrest, presenting rhythm) were summarized using frequencies and percentages.

Compliance with the predefined audit criteria was calculated and expressed as percentages to illustrate performance against standards. Analysis of assessment completion rates distinguished between pre-discharge and post-discharge phases, with additional stratification by assessment type (cognitive screening, emotional problems, fatigue).

Comparative analyses between centres were not performed as the audit’s main objective was to highlight practice variations collectively and generate shared learning rather than evaluate individual centre performance.

All statistical analyses were performed using SPSS version 29.0.1.

## 3. Results

A total of 143 OHCA patients were discharged alive from hospital across the 4 sites during the 6 months of this audit. Descriptive variables are presented in Figure 2. Normality testing using the Kolmogorov–Smirnov test indicated that time to ROSC, length of hospital stay, days from inpatient assessment to hospital discharge, and days from hospital discharge to follow-up were not normally distributed (*p* < 0.001), whereas the assumption of normality could not be rejected for age (*p* = 0.2) (Figure 2).

Adherence to pre-discharge criteria (Criterion A) was assessed in the full cohort (N = 143), while adherence to follow-up content criteria (Criterion B–E) was assessed among those who attended follow-up (N = 87).

Pre-discharge functional assessments meeting our operational definition were completed for 81/143 patients (56%). In a further 35 cases, an assessment was attempted but not completed because patients declined (*n* = 3), were discharged before the assessment could be administered (*n* = 12), the assessment was initiated but not finished (*n* = 5), or they were unable to participate due to severe disability (modified Rankin Scale > 3; *n* = 15). In 27 cases, clinicians did not administer a performance-based functional assessment as defined in Table 1.

For follow-up, the audit tracked provision of information and support. Of the 143 patients discharged alive, 11 died before follow-up, 24 were not invited because of local policies (discharged out of area, not admitted to ICU), and 21 had follow-up arranged but not completed (non-attendance or no contact). The remaining 87 (61%) attended at least one structured follow-up. Complete discharge and follow-up dates were available for 76/87 patients—unfortunately, a system-wide change in electronic records in one of the sites in October 2023 led to missing follow-up date entries for 11 patients. Among the 76 with confirmed dates, only 19 (25%) were seen within the recommended three months, with a median follow-up time of 185 days [IQR 81–225] [Figure 3].

Regarding follow-up content, cognitive screening was completed in 44%, screening for emotional problems in 52%, and fatigue assessment in 51%. The most common reason for omission was patients declining standardised assessments after reporting no problems; other reasons were clinician discretion, time constraints, or severe cognitive impairment [Figure 4; panel 3B]. A family member was invited in all cases, but only 39/87 (45%) attended.

## 4. Discussion

In this prospective audit we investigated compliance with post-resuscitation ERC guidelines across four tertiary cardiac arrest centres in the southeast of England that were, at the time, providing both in-patient and follow-up care. Guidelines were operationalised into five measurable criteria, and compliance was evaluated by collecting additional information on implementation barriers when a standard could not be followed or was not adhered to. We also reported the percentages of assessments attempted but ultimately unsuccessful, with a view to providing insights into real-life implementation challenges. An infographic overview of these criteria, care pathways and observed implementation gaps is provided in Figure 4. In this discussion, we classify barriers using the Theoretical Domains Framework—an established taxonomy in implementation research that helps identify reasons why things do not happen in practice.

### 4.1. Pre-Discharge Care

In terms of pre-discharge care, the discrepancy between completed and offered assessments was due mainly to four factors: patients’ level of cognitive impairment, patients declining the assessment, clinicians not having enough time to complete them, and assessments not being offered as existing information was considered sufficient.

In this cohort, a poor neurological outcome was relatively uncommon—only 15/143 (9.5%) of patients showed marked neurological impairment; this is in line with data from Western countries, where withdrawal of life sustaining treatment is routinely practiced [15]. Nonetheless, this subgroup has highly complex rehabilitation and information needs that are best met by access to specialist neurorehabilitation services, as advocated in the NICE Guideline NG211 (“Rehabilitation after traumatic injury”) and the Royal College of Physicians guideline on prolonged disorders of consciousness. In the larger group of survivors with less significant neurological and cognitive problems, the second, third and fourth factor can be mapped, respectively, to the “Beliefs about consequences (patient-related)”, “Environmental context and resources” and “Beliefs about consequences (clinician-related)” TDF domain. Whilst the former was a minor barrier in this audit, the second affected a larger proportion of patients. Suggested mitigation measures include adding a checklist item to the discharge summary, flagging patients at risk of being discharged early, and/or increasing capacity in the system by providing additional administration or clinician time (Table 2).

In cases coded as ‘not offered’, free-text comments and discussions with participating teams suggested that this label was often used when acute time pressures coincided with reassuring information from standardised cognitive tests, mobility assessments and ward observations, and an additional functional assessment was judged unlikely to change management for that admission. Embedding functional assessments within local standard operating procedures and protecting sufficient clinical time and staffing to complete them may therefore be key to ensuring they are used consistently, particularly given their potential to uncover higher-level difficulties that may be missed by standardised tests alone.

### 4.2. Post-Discharge Care

Inconsistent provision of follow-up care was due to several factors, largely mapping to three TDF domains—namely “Environmental context and resources”, “Beliefs about consequences/motivation (patient-related)” and “Social Influences”.

From an environmental perspective, local organisational policies played a central role. At several sites, operational procedures did not mandate follow-up for patients not admitted to ICU or discharged outside the hospital’s catchment area. As a result, many survivors were never offered a follow-up appointment. Even among those who were invited, specific assessments were sometimes omitted because they were not embedded in the routine follow-up workflow and remained discretionary (e.g., 15, 8 and 4 patients not offered cognitive, mood and fatigue assessments, respectively). Limited clinic time further restricted delivery (6, 5 and 4 patients where cognitive, mood and fatigue assessments were planned but not completed). Overall capacity constraints were also reflected in timing: only one quarter of patients with complete data were seen within the recommended 90 days, with a median follow-up around six months after discharge. Patient-related beliefs and motivation also appeared influential. Among those offered an appointment, around one fifth did not attend or could not be contacted, and among attendees a sizeable minority declined individual components of the assessment (standardised cognitive assessment, mood or fatigue screening). This suggests that some survivors either did not perceive a likely benefit from further assessment or were reluctant to take on additional healthcare burden. However, such decisions are unlikely to be purely individual: cultural norms, stigma, language barriers, family support and practical or financial constraints may all influence whether patients engage with follow-up, mapping onto the “Social influences” domain. Similar factors probably contributed to the relatively low attendance of relatives: although family members were invited for all patients seen in clinic (N = 87), fewer than half (39; 45%) attended, and none of the centres had procedures for actively involving relatives when the patient themselves declined or could not be reached. Another important consideration, although not directly captured in our audit data, is that many survivors already have scheduled contact with cardiology services for secondary prevention (for example, PCI follow-up, ICD clinics and cardiac rehabilitation programmes). At present, these pathways often run in parallel to survivorship assessments, representing a missed opportunity. Future service development and audit work should therefore explore better integration between survivorship pathways and these established secondary-prevention clinics, so that organisational policies and follow-up structures facilitate post–cardiac arrest care.

## 5. Limitations

This audit has several limitations. First, it was observational and non-comparative in design. While it allowed for the prospective collection of implementation data across multiple centres, it did not aim to evaluate the effectiveness of specific interventions, compare performance between sites, or assess clinical outcomes. As such, findings should be interpreted as descriptive of current practice rather than indicative of impact.

Second, the lack of standardisation across centres limited internal consistency. Variations in how assessments were administered, documented, or prioritised reflected local resource constraints and existing workflows, making direct comparisons between sites inappropriate. This heterogeneity is informative but constrains generalisability.

Third, selection bias may have influenced follow-up data. Although 132 patients were eligible for follow-up, only 108 were invited, with some patients excluded due to local referral procedures—for example, those not admitted to ICU or discharged outside a hospital’s catchment area. This may have skewed findings toward those more likely to engage with services.

Fourth, while the audit captured which tools were used, it did not assess how consistently or accurately they were applied, or whether staff received training. The psychometric robustness of these tools in this context was also not examined, and some are not specifically recommended for OHCA survivors.

Finally, the audit did not include any formal measures of acceptability, burden, or perceived utility of assessments from the perspectives of patients, relatives, or clinicians. These insights are critical for guiding meaningful quality improvement and ensuring that interventions are not only deliverable but valued by those involved.

In addition, this audit was conducted at four large hospital trusts that already had infrastructure in place to offer post-resuscitation follow-up care and had expressed an interest in improving practice. As such, these centres likely represent a best-case scenario within the wider system. The level of care delivered across all trusts nationally—many of which may lack structured follow-up pathways or access to rehabilitation staff—is likely to be lower. This selection bias should be considered when interpreting the findings and planning for wider implementation.

## 6. Conclusions

This multicentre prospective audit provides insights into the current implementation of post-resuscitation ERC guidelines across four specialist centres in southeast England. By operationalising guidelines into measurable criteria, we identified specific areas of incomplete adherence and, crucially, barriers mapped to the Theoretical Domains Framework (TDF).

Barriers related to Environmental context and resources were particularly prominent, notably affecting timely assessments both pre-discharge and during follow-up. Limited staff availability, short hospital stays and fragmented or inconsistent follow-up procedures indicate the need for organisational strategies such as standardising assessment workflows, clarifying responsibilities, embedding assessment tasks into routine discharge summaries, and ensuring sufficient clinical and administrative capacity.

Patient-related factors mapped onto the domain of Beliefs about consequences, also significantly influenced guideline adherence. Patients and relatives frequently declined or did not engage with offered assessments, highlighting a need for approaches that enhance their understanding of the value of follow-up and reduce perceived burden, such as framing assessments as routine care, providing early informational interventions, or offering flexible attendance formats (e.g., telehealth options).

Future improvement efforts should leverage these TDF insights, prioritizing interventions that directly target the identified behavioural domains. This could include operational changes (e.g., protected time for follow-up assessments, integration with established secondary-prevention clinics), targeted training and education programs to shift clinician perceptions and behaviours, and improved patient and relative communication strategies [Appendix A]. Adopting a structured, theory-informed approach offers the greatest potential to enhance guideline adherence, ensuring comprehensive, equitable, and sustainable rehabilitation care following cardiac arrest.

## Figures and Tables

**Figure 1 jcm-15-00174-f001:**
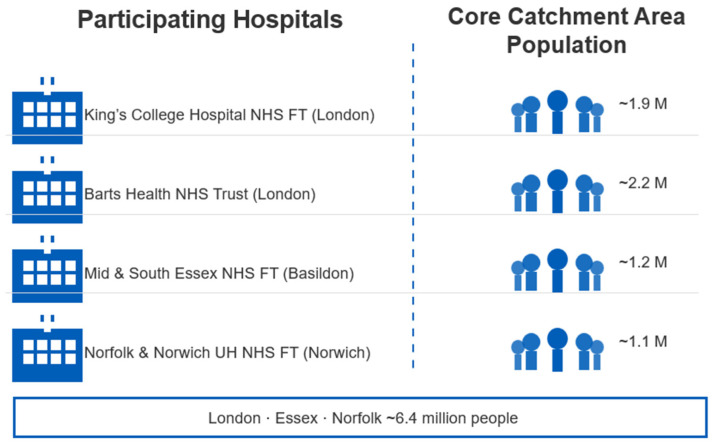
Hospitals taking part to the audit and their relative core catchment area.

**Figure 2 jcm-15-00174-f002:**
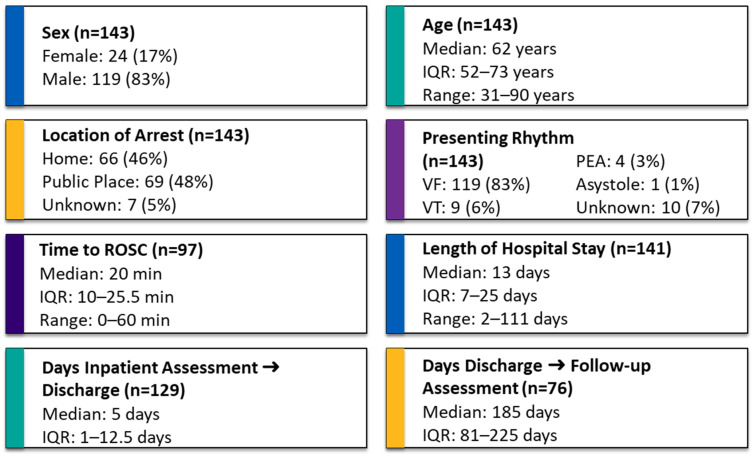
Summary statistics are reported as median, interquartile range (IQR), and range for continuous variables, and as counts and percentages for categorical variables. Subgroup sizes (*n*) are specified where data were incomplete.

**Figure 3 jcm-15-00174-f003:**
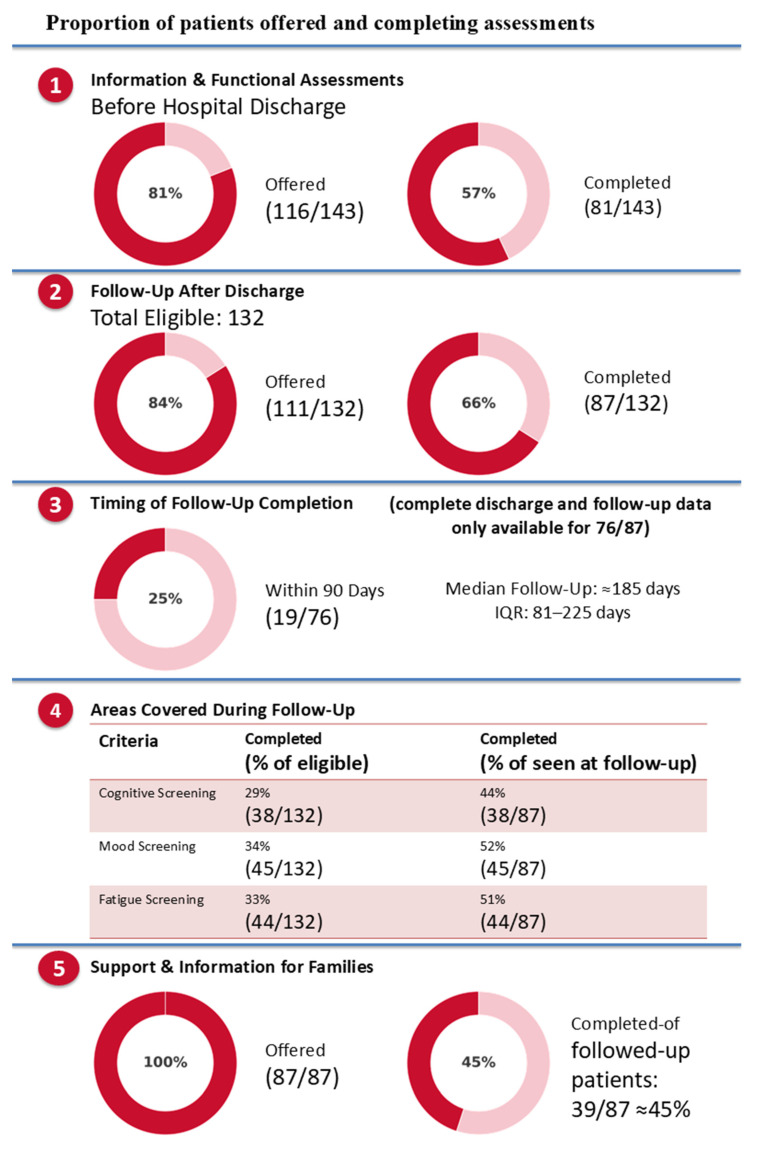
Section 1 shows the proportion of patients offered and completing information and functional assessments before hospital discharge; Section 2 the percentage of eligible survivors who were offered and completed follow-up after discharge; Section 3 reports the timing of follow-up completion, highlighting how many were seen within 90 days and the median follow-up time; Section 4 details the content of follow-up, showing the proportion of patients who received cognitive, mood, and fatigue screening; Section 5 presents the extent to which survivors’ families were offered and completed follow-up support.

**Figure 4 jcm-15-00174-f004:**
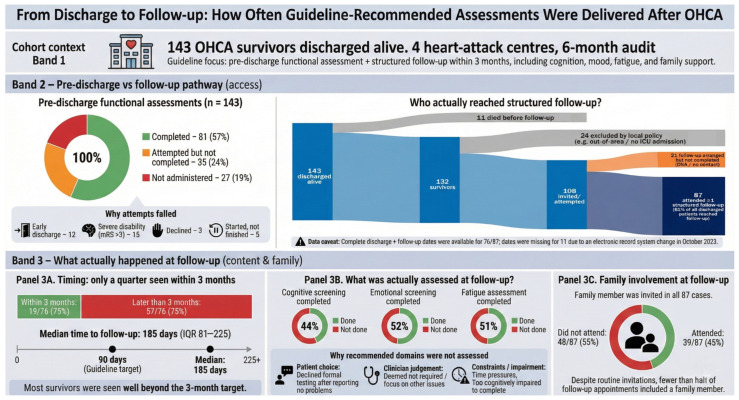
Infographic summarising a 6-month, four-centre clinical audit of 143 out-of-hospital cardiac arrest survivors discharged alive. Band 1 provides cohort context. Band 2 contrasts completion of pre-discharge functional assessments with access to structured follow-up: 56% (81/143) received a pre-discharge functional assessment, 24% (35/143) had an attempted but incomplete assessment, and 19% (27/143) were not assessed; at follow-up, 87/143 patients (61%) attended at least one structured follow-up visit, with others excluded by death, local policy (out-of-area or no ICU admission) or non-attendance. Band 3 shows what happened at follow-up: 19/76 patients with complete timing data (25%) were seen within the guideline-recommended 3-month window (median 185 days after discharge, IQR 81–225); among the 87 completed follow-ups, guideline-recommended screening was performed for cognition in 44%, emotional problems in 52%, and fatigue in 51%, and a family member attended in 45% of appointments despite routine invitations. Timing data were unavailable for 11 patients with completed follow-up because of an electronic records system change.

**Table 1 jcm-15-00174-t001:** Operationalization of audit criteria covering the survivorship pathway. A = pre-hospital discharge assessments; B–E = post-discharge assessments.

Criterion	Operational Definition	Information Recorded
**A. Pre-discharge provision of information and functional assessment**	Performance-based assessment of everyday function (e.g., Multiple Errands Test, kitchen assessment, performance-based ADL assessment) documented before discharge. Cognitive screens, mobility grading, or informal notes alone not included. Documented provision of written or verbal information about cognitive, emotional, and fatigue sequelae, or referral to cardiac rehabilitation with documented education component.	Offered: Yes/No Completed: Yes/no (% of Eligible)
**B. Systematic follow-up post discharge**	Patient was offered a dedicated follow-up appointment or clinic specifically addressing post-arrest sequelae (not routine cardiology review alone), regardless of attendance.	Offered: Yes/No Completed: Yes/no (% of Eligible)
**C. Follow-up completed <3 months**	Days from date of discharge to date of follow-up <90	N seen within 90 days/eligible
**D. Follow-up covers cognition, mood and fatigue**	For each domain, completion of at least one screening tool—e.g., MoCA, ACE-III, FreeCog for cognition; HADS, PHQ-9, GAD-7 for mood; and MFIS/FAS for fatigue.	For each domain: Offered: Yes/No Completed: Yes/no (% of Eligible)
**E. Information and support for families**	At follow-up, documented assessment of the following: cognition, mood/anxiety, fatigue, quality of life, PTSD symptoms.	Offered: Yes/No Completed: Yes/no (% of Eligible)

**Table 2 jcm-15-00174-t002:** Key barriers to completing assessments before discharge and during follow-up after OHCA, aligned with stages of care and mapped to Theoretical Domains Framework (TDF) domains.

Barrier	Stage	TDF Domain	Explanation	Possible Solutions
**Assessment not completed due to discharge/timing**	Pre-discharge	Environmental context and resources	Time pressure, short stays, limited staff	Add checklist item to discharge summary; Flag patients with early discharge risk
**Not attempted—existing information considered sufficient**	Pre-discharge	Beliefs about consequences (clinician-related)	Existing clinical information judged sufficient for discharge decision-making	Clarify Standard Operating Procedures (SOPs); Protect clinical time/staffing
**Follow-up not offered**	Follow-up	Environmental context and resources	Local procedures excluded patients not admitted to ICU or outside catchment area	Review inclusion criteria in SOPs; Introduce alternative follow-up models for excluded groups
**Elements of follow-up assessment not offered (i.e., cognitive ax; mood/fatigue screen)**	Follow-up	Environmental context and resources	Assessment not systematically integrated into follow-up workflow	Identify operational bottlenecks; Clarify SOPs
**Not enough time in follow-up clinic**	Follow-up	Environmental context and resources	Follow-up contact too brief, staff prioritised other elements. Not enough follow-up slots	Streamline assessment battery; Allocate protected time; Create additional capacity
**Patient not contactable or did not attend (DNA)**	Follow-up	Beliefs about consequences/Motivation (patient-related)	Patients could not be reached or did not attend scheduled appointments	Offer reminder calls or texts; Explain relevance of follow-up care before discharge; Offer flexible formats (e.g., phone/video)
**Relative did not attend follow-up**	Follow-up	Beliefs about consequences/Motivation (patient-related)	Relatives may not perceive benefit or may fear burden of involvement	Provide clear invitation for relatives; Explain relevance of family involvement; Offer flexible formats (e.g., phone/video)
**Socioeconomic/cultural reticence in engaging with assessments**	Follow-up	Social Influences	Patients may decline or avoid follow-up assessments due to cultural norms, social stigma, lack of family support, language barriers, or financial/practical difficulties that influence their willingness to engage.	Provide culturally tailored information and materials; Involve community leaders or peer supporters; Offer interpreters and flexible appointment formats (phone/video); Address practical barriers (e.g., transport support, reminder calls).
**Patient declined**	Both	Beliefs about consequences/ Motivation (patient-related)	Patients did not perceive benefit or feared burden	Brief pre-discharge education; Motivational prompts

## Data Availability

The aggregated, anonymised audit dataset underlying this study is available from the corresponding author upon reasonable request and subject to local information governance approvals. No publicly archived datasets were generated.

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
