# Peer review of "Translating Guidelines into Practice: A Multicentre Audit of the Implementation of ERC Survivorship and Follow-Up Recommendations After Cardiac Arrest"

_jcm, 2025, doi:10.3390/jcm15010174_

Round 1
Reviewer 1 Report
Comments and Suggestions for Authors
This multicentre audit addresses an important gap in clinical practice: how ERC survivorship and follow-up recommendations after cardiac arrest are implemented in real-world UK centres. Your aim is highly relevant, and the motivation for the study is strong. However, the manuscript as currently presented has several major methodological and reporting limitations that significantly limit its clarity, interpretability, and scientific value.
Methodological transparency is insufficient. Audit criteria are not rigorously operationalized. Lack of standardized data collection procedure across centres. TDF mapping is not methodologically transparent. Follow-up timing varied widely. Lack of a rigorous CONSORT-style summary. Presentation lacks rigor.
Reviewer 2 Report
Comments and Suggestions for Authors
Thank you for the opportunity to review this paper.
This prospective multicenter audit including 143 OHCA survivors from four tertiary cadiac arrest centers provide important conclusions and future directions in this field with no previous studies and a lack of data about the implementation of the 2021 ERC guidelines (2025 updated). The methods are adequately described, the results were clearely presented by the authors.
Minor comments:
- The results of this study show that only 9.5% of patients had a poor neurological outcome, with the majority of the study population having less significant neurological and cognitive problems. Please add more data, if available, and comments about the secondary prevention procedures and their impact on the pre-discharge (i.e urgent coronary reperfusion-PCI) or during follow-up (i.e ICD implant for SCD secondary prevention). Is there an impact of these intervention on the barriers described in this study (Table 2): the fatigue , mood-anxiety of these patients, the more active involvement of their relative/families in the follow-up process?
- Based on the study results, the SOP procedures and the existing guidelines, please add a comment about a larger team (as future direction) with a crucial role in the first 3 months post-discharge, including the clinician specialist for the evaluation of fatigue/mood-anxiety/cognitive status and their improvement and the specialists involved in the follow-up of the patients who underwent to interventions (i.e. PCI, ICD with rigurous follow-up timeline and enhanced comlience of patients and their families) in order to improve the percentages presented in this study (40-45-50%)
Reviewer 3 Report
Comments and Suggestions for Authors
Dear Authors,
This is a very well structured scientific paper, which could impact the postresuscitation evaluation strategy.
For the best intePlease review the following items to enhance the clarity of the paper:
1. Figure 3 (Section 3) indicates that only 25% (19/76) of follow-ups were completed within 90 days. The denominator used is N=76. The text should explicitly clarify why N=76 was utilized instead of the total number of survivors who completed follow-up (N=87). This discrepancy of eleven patients requires clarification for accurate interpretation of the timing criterion.
2. Improve the placement of citations within the Abstract..
3.To enhance the paper's impact, a more robust emphasis in both the Abstract and Conclusion on the substantial patient exclusions resulting from local policy, specifically regarding catchment area and ICU admission, would be beneficial.
4. The pre-discharge assessment standard references a "functional assessment of physical and non-physical impairment." The text indicates that these assessments are frequently unstandardized and informal in routine practice. Considering that 27 cases relied on "other formal or informal observations... rather than on functional assessment, which were therefore not completed," it is crucial to clarify the operational definition employed by the audit team for a completed functional assessment (Criterion A) to ensure reproducibility and consistency.
Round 2
Reviewer 1 Report
Comments and Suggestions for Authors
No.